# Longitudinal development of sex differences in the limbic system is associated with age, puberty and mental health

Gloria Matte Bon [1,2] ✉, Jonas Walther [1], Erika Comasco[2], Birgit Derntl [1,3] & Tobias Kaufmann [1,3,4] ✉

Sex differences in mental health become more evident across adolescence, with a two-fold increase of prevalence of mood disorders in females compared to males. The brain underpinnings remain understudied. Here, we investigated the role of age, puberty and mental health in determining the longitudinal development of sex differences in brain structure. We captured sex differences in limbic and non-limbic structures using machine learning models trained in cross-sectional brain imaging data of 1132 youths, yielding limbic and non-limbic estimates of brain sex. Applied to two independent longitudinal samples (total: 8184 youths), our models revealed pronounced sex differences in brain structure with increasing age. For females, brain sex was sensitive to pubertal development (menarche) over time and, for limbic structures, to mood-related mental health. Our findings highlight the limbic system as a key contributor to the development of sex differences in the brain and the potential of machine learning models for brain sex classification to investigate sex-specific processes relevant to mental health.

While the neurosciences have for long omitted the study or even the descriptive report of sex differences, it becomes increasingly clear that sex differences exist in brain and behavior across the lifespan[1–4]. Sex differences in the brain are present already in early life yet become more evident during childhood and adolescence[5]. It is during that time of neurodevelopment when many of the common mental disorders emerge[6,7]. Interestingly, sex differences also exist in mental health symptoms, prevalence rates and timing of mental disorders[8,9]. For example, depression and anxiety are more common in females while autism spectrum disorder and attention deficit hyperactivity disorder are more common in males[10]. On a behavioral level, girls are more prone to develop internalizing behavior, while externalizing behaviors are more commonly developed by boys[11]. From the timing perspective, sex differences are reported in schizophrenia onset, with females developing symptoms later in life compared to males[9].

Adolescence demarks a period of particular developmental dynamics in the brain, with a strong impact of sex-related processes[12–14]. This transition period between childhood and adulthood entails a series of gonadal hormonal changes associated with pubertal maturation[13,15,16] that implement profound physical and social changes with long-lasting effects on brain and behavior[8,16]. Previous reports have associated adolescence with neural and cognitive changes, consistently reporting reductions in cortical gray matter volumes following an inverted U-shape trajectory, increasing white matter volumes, as well as improved executive control and social cognition from childhood to adolescence[15–17]. The trajectories of such changes have been found to differ between sexes[16]. Interestingly, there are also sex differences in the timing of puberty onset[13,17], with girls usually developing 1–2 years earlier than boys. Neuroimaging findings of sex differences in neurodevelopmental trajectories map onto such differences in timing of puberty onset, with girls showing earlier peaks in structural changes compared to boys[8,15,16,18,19]. However, a lack of longitudinal studies and precise measures of pubertal development have made it difficult in the past to disentangle the effect of age from those of puberty when studying neurodevelopment[16]. Brain development is a long-lasting process that starts in early life and continues until adulthood, characterized by the interplay of genetic, hormonal and social factors. In this framework, age and puberty, highly interconnected, might interact in establishing sex differences in the brain, by differentially affecting the temporal dynamics of brain development across sexes[20,21]. While certain manifestations of sex differences in the brain may be directly attributable to

[1]Department of Psychiatry and Psychotherapy, Tübingen Center for Mental Health, University of Tübingen, Tübingen, Germany. [2]Department of Women's and Children's Health, Science for Life Laboratory, Uppsala University, Uppsala, Sweden. [3]German Center for Mental Health (DZPG), Partner Site Tübingen, Tübingen, Germany. [4]Centre for Precision Psychiatry, Division of Mental Health and Addiction, Institute of Clinical Medicine, University of Oslo, Oslo, Norway. ✉e-mail: gloria.matte-bon@uni-tuebingen.de; tobias.kaufmann@med.uni-tuebingen.de

the biological processes underlying pubertal development, others may be attributable to age dependent social or genetic factors that go beyond puberty. As such, disentangling the effects of age and puberty is crucial to fully understand brain dynamics across adolescence.

While sex has been traditionally considered as binary in research, there is an ongoing debate to what degree sex differences in the brain may deviate from a binary distinction[22–24]. It has been suggested that sex differences may run on a male-female continuum, where individuals can even manifest on different parts of this continuum depending on the brain region under investigation[22]. Machine learning models that classify individuals according to their biological sex assigned at birth based on brain images offer a window into a brain sex continuum[25]. The models not only return a binary class label but also provide a probability for that label, essentially generating for each individual person an estimate of position on a continuum that runs from a male-like to a female-like brain. Importantly, such position is not an expression of gender identity, but rather a manifestation of the overall pattern of sex differences across the brain regions under consideration. This approach has been used in the literature to study the brain sex continuum in both adolescents and adults[5,11,26], pointing toward higher sensitivity of the continuous estimates to capturing sex-related biological associations[11]. Kurth and colleagues[5] used structural imaging phenotypes to investigate changes in brain sex classification accuracy in association with age across childhood and adolescence, showing better performance for adolescents compared to children and increased magnitude of sex differences with increasing age. Leveraging similar approaches, Vosberg and colleagues[11] showed that a continuous sex-score based on brain and body was associated with hormonal and behavioral phenotypes. However, the models underlying brain sex estimates are usually based on whole-brain measures, potentially overlooking regional-specific effects.

Recently, we developed a regionally constrained machine learning model for brain sex based only on limbic structures[27]. The limbic system is a key player in emotional processing and regulation, functions highly affected in many mental disorders. Moreover, limbic structures have high expression of receptors for gonadal hormones and have been shown to go through changes across hormonal transition periods in females[17], including puberty. In fact, pubertal changes in limbic structures have previously been reported[8,13,18], suggesting the limbic system as potential target for the study of sex differences during pubertal development.

Here, we investigated the potential of regionally constrained limbic models for brain sex classification to unveil sex-specific effects of pubertal development on brain structure. To this end, we first used neuroimaging data of children and adolescents from two cross-sectional samples - the Philadelphia Neurodevelopmental Cohort (PNC)[28] and the Human Connectome Project Development (HCP-D)[29] - to train machine learning models for brain sex classification based solely on limbic or non-limbic volumes, and compared them to models trained on the whole brain. We then externally validated our models in two large longitudinal datasets, the Queensland Twin Adolescent Brain (QTAB)[30–32] and the Adolescent Brain Cognitive Development (ABCD)[33]. For all included samples, we investigated whether the resulting class probabilities for each model are associated with age. Our hypothesis was that under the influence of age and pubertal maturation, the sex differences in the brain will increase, allowing the models to better classify subjects based on their biological sex at follow-up compared to baseline, and that this increase will be particularly significant for the limbic model given the relevance of the limbic system for pubertal maturation. We then investigated the impact of pubertal development on this process within each sex in QTAB and ABCD data. We expected sex-specific associations of pubertal development with class probabilities, with stronger association for the limbic model. Finally, we investigated the association with mental health measures for anxiety and depression in the QTAB sample, again expecting to find stronger effects for the limbic model in females.

## Results

### Brain sex model performance increases longitudinally in youths

Female-male differences in total brain volume exist, which is not the variability of interest that our models aimed to capture. Therefore, we implemented a pipeline[34] that matched females and males in the training sample on estimated total intracranial volume (eTIV), in addition to age and Euler number (a proxy measure of image quality[35]), yielding a balanced sample with a 1:1 ratio females to males ($N = 1132$, age: 8–22 years, 50% females, cross-sectional PNC and HCP-D samples). In addition, we regressed eTIV from all brain features prior to training machine learning models on limbic, non-limbic and whole brain data, respectively.

All three machine learning models achieved solid sex classification performance, with an area under the curve (AUC) from 5-fold cross validation of 0.79 (limbic), 0.76 (non-limbic) and 0.82 (whole brain). Figure 1A depicts the corresponding receiver operating characteristic (ROC) curves. It is worth noting that the limbic model yields performances comparable to the other models with much less features (limbic: 160 features; non-limbic: 333 features; whole brain: 493 features). Interestingly, when comparing the feature contributions in the two regionally constrained models to the feature contribution in the whole brain model, we found a considerable overlap between the whole brain and respective regional models. Similar features contributed to the regional and whole brain models, indicating that the model predictions are robust (Supplementary Fig. S1). Of note, eight out of the first ten features contributing to the whole brain model were limbic features, providing further evidence of the relevance of limbic features to uncover sex-specific factors.

When testing the model in two independent longitudinal samples (QTAB: $N_{Baseline} = 392$, age at baseline: 9–14 years; and ABCD: $N_{Baseline} = 7792$, 47% females, age at baseline: 9-11 years), we obtained overall lower but still high performances (Fig. 1B). Interestingly, both datasets showed a longitudinal increase in accuracies at 2 years follow-up compared to baseline for all models (QTAB: limbic: 0.68 (baseline) → 0.70 (follow-up), non-limbic: 0.67 → 0.71, whole brain: 0.72 → 0.75; ABCD: limbic: 0.71 → 0.73, non-limbic: 0.74 → 0.75, whole brain: 0.76 → 0.78), suggesting that the models were able to better classify according to biological sex when the subjects are older.

### Brain sex class probabilities are associated with age in a sex-specific manner

Given the longitudinal performance increase, we tested for age associations with the class probabilities. Since the better performance with increasing age is expected to manifest as a shift of the class probabilities toward the opposite extreme of the distribution for each sex, resulting in an increase in class probabilities toward more female-like brain for females (binary coded in the model as 1), and a decrease toward more male-like brain for males (coded in the model as 0), we carried out the analysis within each sex separately.

In the training set, we found that for females increasing age was significantly associated with an increase in class probabilities (i.e. more female-like brain) for all models (Fig. 1C), with the strongest effect for the non-limbic model (limbic: $t = 2.399$, $p = 0.01676$; non-limbic: $t = 7.485$, $p = 2.79 \times 10^{-13}$; whole brain: $t = 4.213$, $p = 2.94 \times 10^{-5}$; all models accounted for Euler number and site). In males, no significant association was found for the limbic ($t = -0.756$, $p = 0.450024$) and whole brain ($t = 0.391$, $p = 0.696$) models, while a significant association in the direction of more female-like brain was found for the non-limbic model ($t = 3.501$, $p = 0.000501$) (Fig. 1D).

Next, we tested if similar age effects as observed in the cross-sectional training set were also visible in longitudinal data. In the two longitudinal test samples, we applied a Linear Mixed Effects (LME) model, accounting for both age and session and controlling for site and Euler number as further covariates. Figure 1E shows the F-values for the main variables of interests (age, session and the interaction between the two) for all models in both sexes and datasets. Across both samples, age associations with class probabilities of the limbic and whole brain model were much stronger in females than males. Furthermore, a general pattern of stronger effects for the limbic compared to the non-limbic model was found across samples and sexes, with the exception of males in QTAB showing significant associations only in the non-limbic model (Fig. 1D, E). In addition, in ABCD in males a significant interaction age

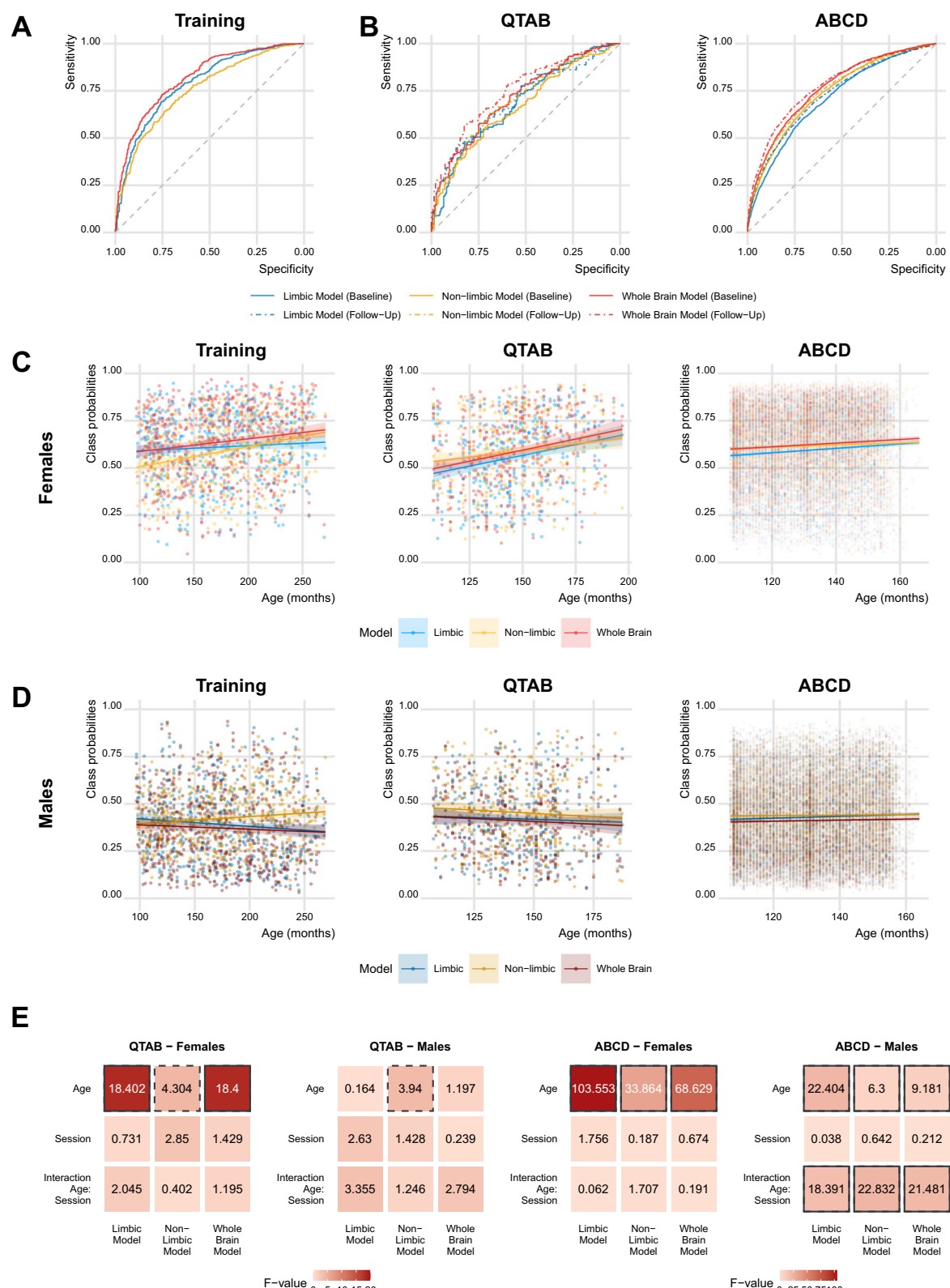

by session was found for all models, with an increase in class probabilities at follow up compared to baseline (Fig. 1D, E).

### Limbic models are most sensitive to pubertal development

Both longitudinal samples – ABCD and QTAB - allowed us to investigate the longitudinal association between pubertal development and brain sex. To this end, we used LME models in combination with ANOVA type I, considering pubertal development variables as the main effect of interest, while accounting for age, Euler number and site (when applicable) as covariates. Additional analyses using ANOVA type II, which allowed us to quantify the relative contribution of pubertal development vs. age are provided in the next section.

**Fig. 1 | Biological sex can be classified from all three machine learning models with increasing accuracy as individuals get older. A** Model performance in the training sample showed high performances for all three models. **B** External validation of the models in two independent longitudinal samples showed increased performance at follow up compared to baseline. **C** In females, age was associated with an increase in class probabilities for all models and samples, in accordance with the increasing performance observed across sessions. **D** In males, the age association was overall weaker compared to what was observed for females. **E** Linear mixed effect (LME) models showed that the limbic model in females was particularly sensitive to age in both longitudinal validation sets. The maps depict F statistics of the main

variables of interest (confound factors, Euler number, and site were modeled but are not visualized). Solid borders denote significant values after correction for multiple comparisons, dashed borders highlight nominally significant values that do not survive multiple comparison correction. Corresponding plots depicting the p-values for each association are presented in the Supplementary File, Figure S2. Note: **C**, **D** contain two data points per subject for ABCD and QTAB samples and regression lines with 95% confidence interval across all data points are shown. This was done for visualization purposes. Appropriate longitudinal statistics are provided in (**E**).

First, we tested for associations of sex class probabilities with the average Pubertal Development Scale (PDS) score. We used the self-reported children's questionnaires for these analyses for consistency between datasets, due to the wide age range of the QTAB sample (9–14 years at baseline) and the known increased reliability of self-reports compared to caregiver report in older-aged youths[36].

A pattern of stronger PDS association for the limbic model compared to the non-limbic and in females compared to males was found in both datasets. In QTAB, higher PDS scores were significantly associated with higher estimates only for the limbic and whole brain models in females, with stronger effects for the limbic model (Fig. 2A, C), while no significant effect was found in males (Fig. 2B). In the ABCD data, all models except the whole brain model in males showed significant associations with the average PDS score, with stronger effects obtained for the limbic model for both sexes (Fig. 2E–H). Moreover, a pattern of stronger effect for females compared to males was found in both datasets (Fig. 2C, D, G, H), suggesting that in this age range (9-11 years at baseline, 2 years follow-up) the limbic system is able to capture the variance due to pubertal development factors better in females than males. Given the young age range of ABCD, we conducted a control analysis using caregivers' reports, which converged on the same results (Supplementary File, Figure S4).

To understand whether the observed pubertal associations were particularly sensitive to the biological process of menarche, we further tested for the association between menarche onset and class probabilities in females. To this end, we divided our female participants based on the PDS question 5 ("Have you begun to menstruate?") in two groups: 1- participants that did not begin to menstruate at follow-up (answered "No" at both session); 2- participants that begun to menstruate between the two sessions (answered "No" at baseline and "Yes" at follow-up).

In QTAB, LME statistics revealed a significant association between menarche onset and class probability for the limbic and whole brain models, but not for the non-limbic model (Fig. 3A, C). In the ABCD sample, this association was found only for the limbic model, confirming the results obtained in QTAB (Fig. 3B, D). Additional control analyses in ABCD using caregivers' reports showed similar results (Supplementary File, Figure S6).

### Relative contributions of puberty and age
Since age and puberty are strongly intertwined, we further investigate the relative role that each factor plays in the associations with class probabilities by applying an ANOVA type II to the same LME models previously described.

Variance was split between age and measures of pubertal development, with age capturing most of the variance. For PDS associations, only the limbic model in females in the ABCD survived corrections for multiple comparisons (Supplementary File, Figure S7). Likewise, for menarche associations, only the limbic model in ABCD showed an effect, which however did not survive correction for multiple comparisons (Supplementary File, Figure S8).

It is important to point out that the wider age range of QTAB might mask specific effects of pubertal development. To test such hypothesis, we computed the correlation between age and PDS score within each sex for each session separately. Indeed, in QTAB the correlation between age and PDS was highly similar between sessions (Supplementary Fig. S9-A). In contrast, for ABCD the correlation was substantially lower at baseline compared to follow-up (Supplementary Fig. S9-B).

### Sex class probabilities are associated with female mental health
The wider age range of the longitudinal QTAB sample compared to ABCD encompasses more variance in mental health, allowing us for this sample specifically to study the association between class probabilities and mental health. Given our focus on the limbic system and its known relevance for female-prevalent mood-related mental disorders, we studied the children self-report questionnaires on anxiety and depression symptoms. To obtain a general mental health score, we performed a principal component analysis (PCA) for dimensionality reduction at baseline. We selected the first component ($PC1_{baseline}$) as general mental health score across depression and anxiety scales, as it explained 21.5% of the variance and was correlated with all the questions included. To assess the same score at follow-up we used the obtained loadings for each question for the first component and applied them to the follow-up mental health data (Fig. 4A). The obtained $PC1_{follow-up}$ scores were positively correlated with the $PC1_{baseline}$ scores (Fig. 4B, $r = 0.42$, $t = 8.09$, $p = 1.43 \times 10^{-14}$).

The main analysis was conducted using LME in combination with ANOVA type I, assessing the main effect of mental health. In females, we observed significant associations of mental health with the whole brain and limbic class probabilities. No significant association was found for the non-limbic model in females or for any of the models in males (Fig. 4C).

Further analyses to explore the relative contributions of mental health, age and sessions using ANOVA type II showed the same patterns of strongest effects for limbic and whole brain models in females, although only the whole brain model survived correction for multiple comparison (Supplementary File, Figure S11).

### Discussion
This work highlights the potential of limbic models for brain sex to unveil sex-specific effects of puberty on the brain. Sex differences in brain structures emerge already in the prenatal period and increase across childhood and adolescence. Puberty and menarche exert a key role in this process. Such process is mirrored in the emergence of sex differences in mental health prevalence around puberty. Limbic structures, with high concentrations of sex-hormones receptors, represent a good candidate to investigate the development of sex differences across these periods. Here, we demonstrated in two independent samples that regionally constrained machine learning models for brain sex are sensitive to age, pubertal development, and mental health, with the strongest effects in the limbic model. Moreover, a pattern of stronger effects in females compared to males was found for all the analyses, suggesting that these effects are to some degree sex-specific.

The impact of adolescence on brain structures has thus far mostly been studied in an age framework, where previous work has shown sex sensitive age trajectories across various measures of brain anatomy and function[18,19,37,38]. This framework may, however, not be the most sensitive framework to study brain development as it may omit biological variance related to the individual stage of pubertal maturation. Here, we attempted to disentangle different sources of variance, modeling the effects of age, pubertal developmental stage, and the specific process of menarche in two longitudinal data sets. In line with the literature[5], our findings showed a significant association of brain sex with age, indicating a better classification according to the biological sex with increasing age. Interestingly, the age association was stronger in females compared to males, with a pattern of strongest effects for the limbic model across datasets. These findings align

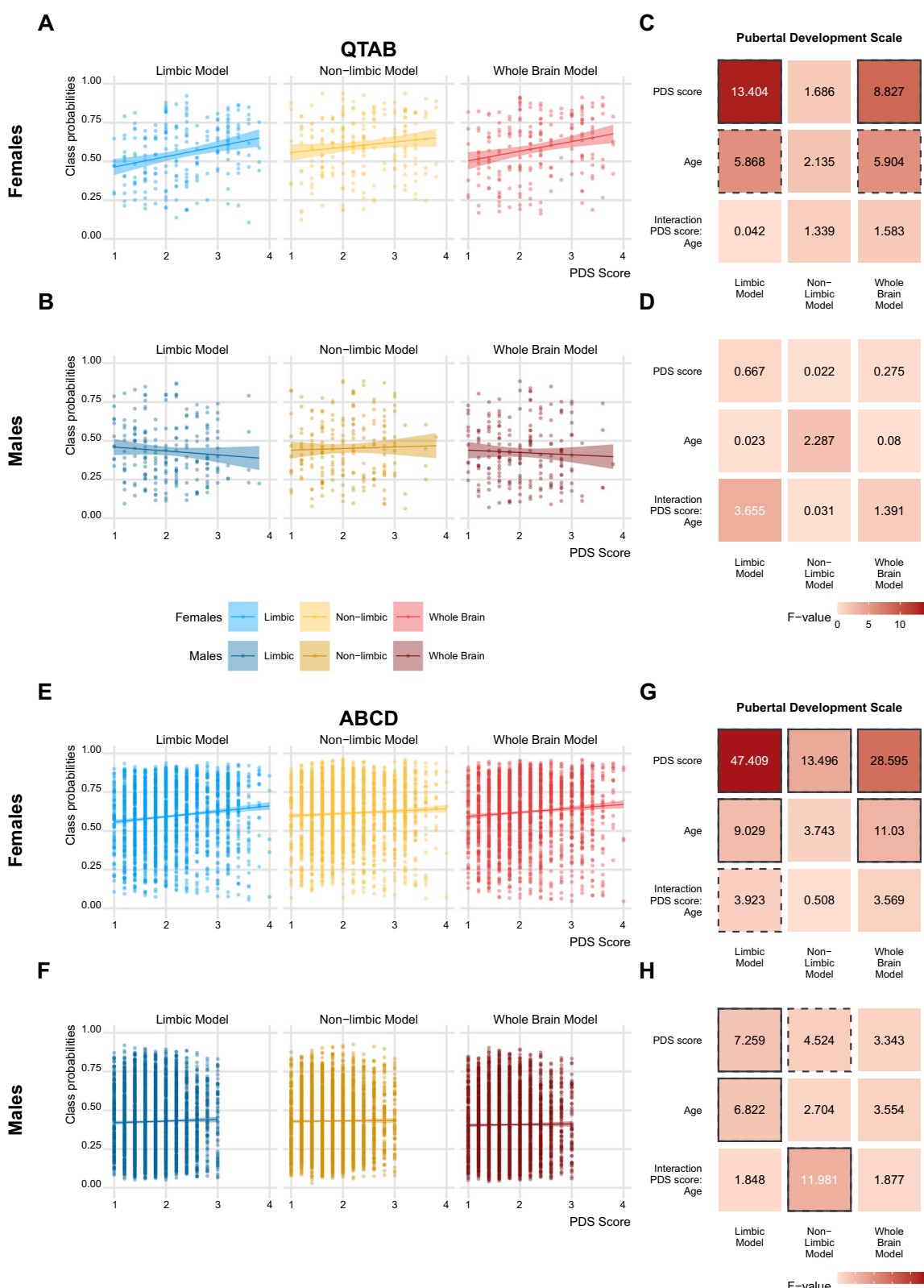

**Fig. 2 | Class probabilities are associated with pubertal development in a sex-specific manner.** In QTAB (**A**–**D**), class probabilities were significantly associated with average PDS score for the limbic and whole brain model in females (**A**), while no significant effect was found in males (**B**). LME models yielded a pattern of stronger effects in the limbic compared to the non-limbic model and in females compared to males (**C**, **D**). In ABCD (**E**–**H**), all models showed significant PDS score association in both sexes, with the exception of the whole brain model in males (**E**, **F**). A pattern of strongest effects in the limbic system was found, as well as a pattern of

stronger association in females compared to males (**G**–**H**). Solid borders denote significant values after correction for multiple comparisons, dashed borders highlight nominally significant values that do not survive multiple comparison correction. Corresponding plots depicting the p values for each association are presented in the Supplementary File, Figure S3. Note: Scatter plots contain two data points per subject samples and regression lines with 95% confidence interval across all data points are shown. This was done for visualization purposes. Appropriate longitudinal statistics are provided in heatmaps.

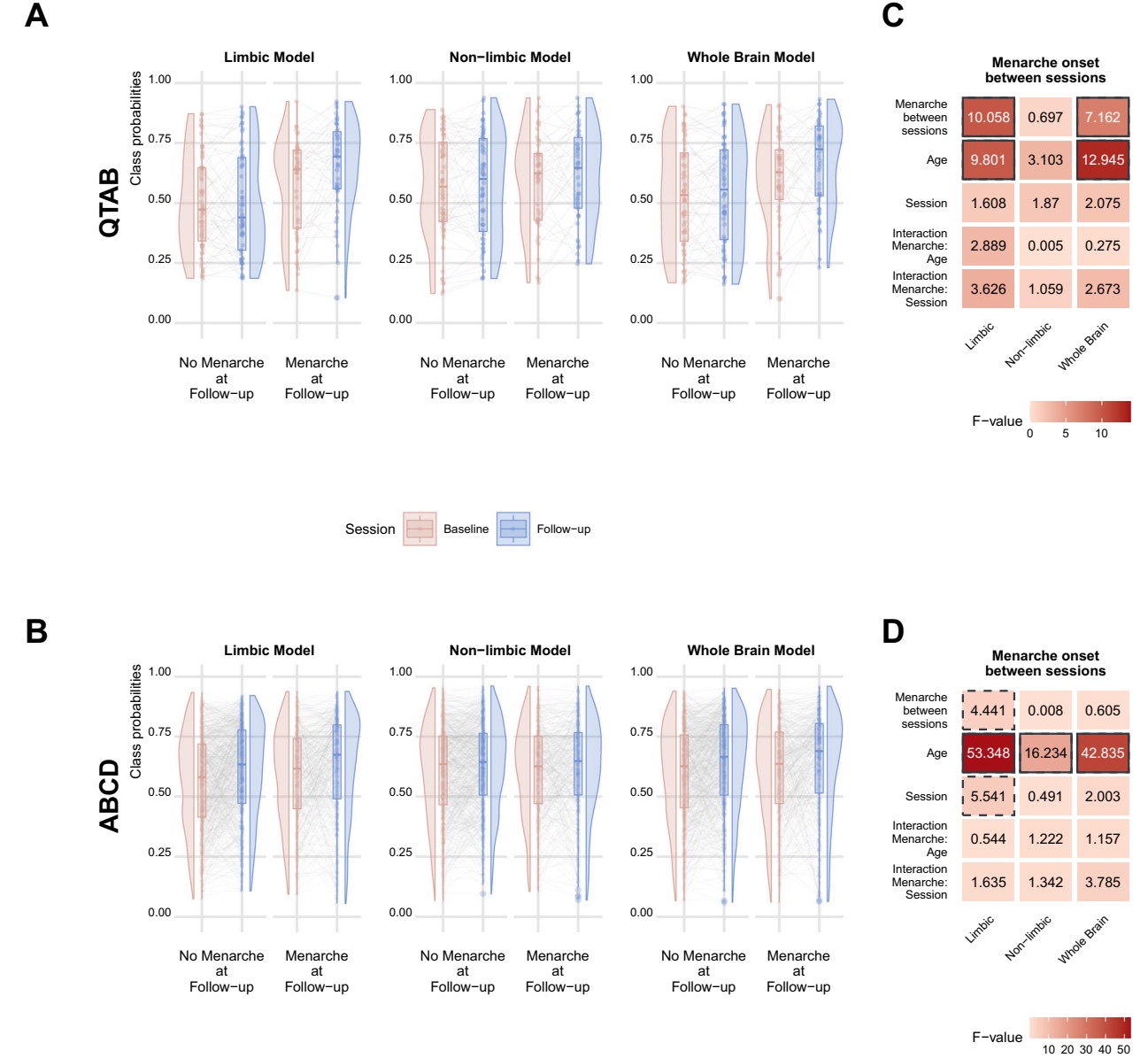

**Fig. 3 | Limbic class probabilities are sensitive to menarche onset. A** In QTAB, class probabilities for the limbic and whole brain but not the non-limbic model were significantly higher in the group that had menarche at follow-up. **B** In ABCD, the same association was significant only for the limbic model. The whiskers in the boxplot extend from the hinge to the largest/smallest value, no further than 1.5 * IQR from the hinge. Heat maps illustrate corresponding LME statistics for QTAB (**C**) and ABCD (**D**). Solid borders denote significant values after correction for multiple comparisons, dashed borders highlight nominally significant values that do not survive multiple comparison correction. Corresponding plots depicting the *p*-values for each association are presented in the Supplementary File, Figure S5.

with a recent brain age study reporting a developmental age-related delay in boys compared to girls in a similar age range[39]. Of note, the structures contributing the most to the brain age model were limbic structures, suggesting greater emotional maturity in females compared to males in that age range[39] and further confirming our hypothesis of the limbic system as a good target to study the developing brain.

The age associations in our study were reflected at the accuracy level in the two longitudinal samples, showing improved performances at follow-up compared to baseline for both regional models and the whole brain model. It is important to highlight that our models were stringently corrected for estimated total intracranial volume (eTIV) differences. We matched females and males in the training sample for eTIV, in addition to regressing out eTIV from the raw features prior to the machine learning procedure. This is an essential step as total brain volume represents a major confounding factor when investigating sex differences. Sex differences in total brain volume have been consistently reported across the lifespan, with males showing

8-15% larger brain volume compared to females[40,41], yet when controlling for these global effects, brain volumes differ much less between sexes[40]. Factors such as body weight and heights strongly influence total brain volume and increase across childhood and adolescence, with total brain size reaching its peak during adolescence when height and weight velocity are also at their maximum[42]. Therefore, stringent correction for eTIV becomes essential in the age range considered, to ensure that the age associations with class probabilities are not determined by differences in brain size.

After analyzing the effects of age alone, we aimed to disentangle the effects of puberty by applying a separate analysis considering the average Pubertal Developmental Scale (PDS) score as the main variable and including age as a covariate. The effects of puberty on the development of the human brain have long been an object of discussion[43]. Mixed results have been reported in the literature on the relation between puberty and functional and structural brain changes, due to differences in data collection and puberty indicators, as well as the complex nature of the interaction between

**Fig. 4 | Class probabilities are associated with mental health in a sex-specific manner.**
**A** Schematic representation of the principal component analysis (PCA) to obtain the general mental health score at both sessions. **B** The first components between the two sessions were significantly correlated across both sexes. **C** The F-values of the LME model showed significant associations only for the limbic and whole brain model in females, with strongest effects for the whole brain. Solid borders denote significant values after correction for multiple comparisons. Corresponding plots depicting the *p*-values for each association are presented in the Supplementary File, Figure S10.

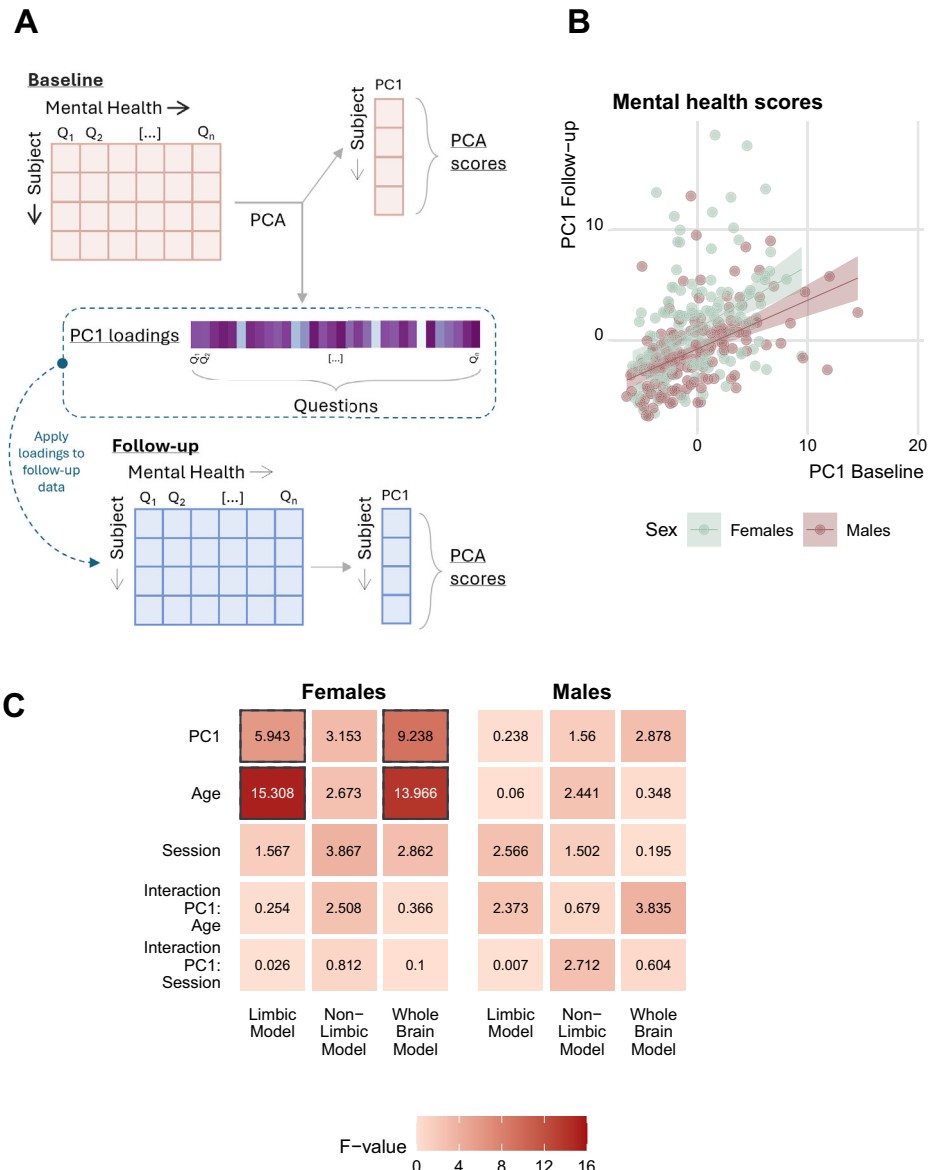

puberty, age and sex[43]. In line with the literature suggesting a differential impact of puberty on different regions in females and males[20,44–46], we found a pattern of stronger associations in females compared to males and in the limbic system across datasets. These effects were further confirmed by supplementary analyses to disentangle the relative contributions of pubertal stage and age in the same model, showing significant associations in the ABCD sample only for the limbic model in females, suggesting puberty as a key player in sexual differentiation for these regions in this age range. Importantly, the lack of significant effects in QTAB for such analysis might be explained by the wider age range of this dataset, showing already strong correlations between age and PDS score at baseline, potentially masking specific contributions of PDS score. This hypothesis is confirmed by studies showing different developmental trajectories in different brain regions in relation to puberty, age and sex[45,46], indicating a better alignment of such trajectories with puberty-related changes such as hormonal levels rather than age-related changes[47]. However, the restricted age range considered (especially in the ABCD sample) might be key in determining the difference in effect sizes between sexes, due to the shift in peak of maturation between boys and girls[46].

The increase in class probabilities toward more female-like brains in females that underwent menarche between the two sessions further support our hypothesis of sex-specific puberty-related changes in brain structure, and align with recent work showing that menarche can be classified from brain structure[48]. The literature regarding the increasing sex hormones during adolescence and sex differences in the brain so far had led to conflicting results[5,12,43]. However, these studies usually focus on few regions of interest, potentially overlooking the overall pattern due to the interplay between structures[5,26,49]. Importantly, our class probabilities are the result of a multivariate classification model that integrates information from different brain regions into a single score. Thus, our approach allows us to tackle the complex interactions that exist between sex, puberty, and age.

Our results on the association of sex class probabilities and mental health for the limbic model in females are in line with our previous findings in adults[27]. Adolescence demarks a pivotal period for the emergence of psychopathology[50], and this is also the time when mental health prevalence rises for females[44,51], suggesting a potential role of steroid hormones, among other psychosocial and environmental factors[50]. Structures of the limbic system, a key player in emotional processing and regulation, have a high concentration of gonadal hormone receptors[17,52]. Importantly, alterations in brain structure and function in adolescents with mood disorders have been frequently reported in the literature[50,53,54]. Together these results confirm our

limbic model as a potential target to investigate sex-specific effects of pubertal development and their relation to mental health outcomes.

The present findings should be interpreted in light of some methodological considerations. Different intracranial volume-correction methods and measures can influence the analysis outcome[55–57]. Here, we adopted a two-step approach to first match subjects based on their eTIV[34], and then regress the eTIV from the raw features prior to the machine learning procedure. While we chose this procedure for its high stringency in addressing eTIV confounds, it must be noted that matching subjects on eTIV may result in a sample that is less representative of the population, i.e. selecting females with the largest volumes and males with the smallest volumes. Through supplemental analyses (Supplementary File, Figure S12), we have ensured that the reference used for matching did not have an impact on the results (taking females as reference to find matching males, or vice versa). We acknowledge that establishing sensible eTIV correction procedures is an ongoing endeavor of the field and future research may yield further developments in this direction.

Furthermore, the use of self-report measures for pubertal development rather than hormonal and physical examinations renders another limitation to this study[43]. While the latter are known to be more sensitive and specific, self-reports have been suggested to be a sufficiently accurate instrument for cohort studies[58]. Nonetheless, self-reported measures of pubertal development have been shown to vary in their accuracy depending on the stage of development[36,59], with caregiver reports being more accurate for younger subjects[36]. While we had youths' self-reports and caregiver reports available, here we decided to use the youths' self-report measures for the main analyses. This decision was based on data availability as well as the wider age range in the QTAB sample and the expected higher accuracy of self-reports for older individuals. To rule out the dependence of results on this selection, we carried out a supplementary analysis in the ABCD sample using the caregiver reports. Importantly, we found even stronger effects when using the caregiver reports, further corroborating our results reported in the main text. Future studies may add to our results by investigating hormonal associations to shed light on the direct effects of puberty-related hormonal changes. Moreover, the ABCD study is an ongoing data collection effort, and future releases will allow the investigation of trajectories into later stages of brain development, including their potential interplay with other socio-cultural aspects, such as gender identity[23,24]. This longitudinal effort will also allow the characterization of mental health effects in ABCD, which we have omitted here due to the relatively low age of the sample and corresponding low mental health prevalences. As such, while the mental health associations that we have observed for limbic sex class probabilities in QTAB match our findings in adults with depression well[27], this aspect warrants further replication in future releases of the ABCD or other independent data sets. Finally, the QTAB samples include twin and sibling participants. For model validation this is not an issue per se as we ensured independence of sex predictions between subjects by predicting at the individual subject level from a model trained in other datasets. However, we did not account for family factors in our LME models when testing for associations of the class probabilities. In light of our previous findings in adults, showing high degree of heritability for class probabilities[27], further research on the impact of kinship in the observed associations might be warranted as we cannot currently quantify the impact of shared genetic factors between siblings.

To conclude, our findings highlight the potential of region-specific models for brain sex to uncover sex-specific processes in human brain development. Our approach showed sex-specific sensitivity to developmental processes, suggesting that leveraging multivariate methods to map sex-related variance in the brain onto female-specific factors may render a promising approach toward the use of sex-sensitive markers in precision psychiatry and, by that, toward more inclusive research.

## Methods
### Participant selection
All ethical regulations relevant to human research participants were followed. Each sample had their study procedures approved by the respective local ethics committees and obtained written informed consent by participants or their caregivers.

For training and cross validation, we used data obtained from the Philadelphia Neurodevelopmental Cohort (PNC)[28] and from the Lifespan Human Connectome Project Development (HCP-D)[29]. After excluding 20 subjects in PNC and 8 subjects in HCP-D that were outliers for the Euler number, an image quality measure, we selected the same age range (8–22 years old) in both datasets, by excluding 19 subjects with an age range of 5 to 7 in the HCP-D. To correct for multiple scanners in image acquisition, we harmonized the volumes across the two dataset and the multiple scan sites in the HCP-D by using the *neuroCombat* harmonization for multi-site imaging data (version 1.0.13)[60]. The harmonization process was run across 5 scan sites (1 site for PNC, 4 site for HCP-D), taking into account sex and age expressed in months as covariates. To obtain a balanced sample for sex while controlling for age and brain size, we applied a matching procedure. We first split the participants according to their sex assigned at birth. Each female was then matched with one male according to the following criteria: 1) maximum absolute age difference of 12 months; 2) maximum absolute difference in the harmonized estimated total intracranial volume (eTIV) of 3%; 3) maximum absolute difference in Euler number of 1 standard deviation. If more than one male fitted the criteria, the one with the lower eTIV difference was selected as the best match. The final sample was composed of $N = 1132$ (50% females; PNC = 768 of which 48% females; HCP-D = 364 of which 53% females), with no significant difference in age (in months), eTIV or Euler number between sexes. Supplementary Table T1 shows the sample characteristic before and after the matching procedure for the selected and excluded participants.

To validate our models independently, we used data from two longitudinal studies: the Queensland Twin Adolescent Brain (QTAB)[30] and the Adolescent Brain Cognitive Development (ABCD)[33]. The QTAB dataset includes adolescent twins ($N_{Baseline} = 392$, 49% females, age at baseline: 9–14 years), scanned twice with an interval of 13 to 30 months (mean interval: 20 months). The ABCD study is a longitudinal study that follows participant throughout adolescence[33]. We used the data release 5.0, including two MRI scans collected 2 years apart, across multiple research study sites ($N_{Baseline} = 7792$, 47% females, age at baseline: 9-11 years). After excluding 15 observations in QTAB and 279 observations in ABCD due to image quality issues, we harmonized the data across sites, using the *LongCombat* package (version 0.0.0.90000), the longitudinal ComBat harmonization for multi-scanner imaging data[61]. Supplementary Table T2 reports the included participants for each analysis.

### Image segmentation and features selection
Raw T1-weigthed MRI scans were preprocessed in FreeSurfer (v. 7.1.1) and the cortical and subcortical reconstruction was carried out for all cross-sectional and longitudinal datasets. Although a longitudinal pipeline is available in FreeSurfer, this approach is discouraged when considering data from children, as it assumes a constant eTIV between sessions. We decided, therefore, to use the traditional pipeline, and account for the longitudinal setting in the harmonization process, as discussed in the previous section. A mix of defaced and non-defaced data was used for both the training (PNC not defaced, HCP-D defaced) and the two independent test samples (QTAB defaced, ABCD not defaced), as provided by the study sources. While defacing may impact brain measures, alignment of results across defaced and not defaced samples supports that findings are robust to potential defacing related issues.

As quality control of the imaging data, we used the Euler number, a proxy of image data quality[35], averaged across the two hemispheres and excluded subjects with values lower than three standard deviations from the mean. The Euler number represents a measure of the topological complexity of the reconstructed cortex and has been shown to provide a useful proxy of structural image quality, given high association with manual rating procedures and outperforming motion indicators in identifying unusable images[35,62]. By excluding outliers for Euler number in each sample we aim to control for MRI quality reconstruction and potential motion artefacts.

For a better fine-tuning and more precise selection of limbic structures, we combined volumes derived from different segmentation approaches, as previously described elsewhere[27]. Briefly, we used the multimodal parcellation of the cerebral cortex[63] together with multiple subcortical segmentation, including structures from the classical subcortical segmentation from FreeSurfer, the segmentation of subcortical limbic structures[64] and the subfield segmentation of the hippocampus[65], amygdala[66] and thalamus[64]. The final feature set comprises 493 features (whole brain segmentation), carefully combined to avoid overlap between features derived with different segmentations. Each feature was then assigned to a limbic (160 features) or non-limbic (333 features) feature set based on the literature definition of the limbic system[64,67–69]. A complete list of limbic features and respective parcellation is listed in the Supplementary table T3, while Supplementary Fig. S13 illustrates the cortical distribution of limbic and non-limbic features.

To correct for differences in head size that could affect the model performance, prior to model training and external validation, we regressed out the eTIV from each feature and took the residuals.

## Sex classification model

We trained one model for each set of features obtained after eTIV regression, using the binary classification implemented in the *xgboost* package (version 1.7.5.1) in R, following routines previously described in Matte Bon et al.[27]. In the training sample, sex was coded as a binary variable (males = 0, females = 1). Briefly, we performed a nested 5-fold cross validation, with the learning rate $\eta = 0.01$ and the initial number of rounds set to 1000. The prediction error in the inner loop was assessed for each iteration and used to determine the optimal number of iterations to train the final models on the entire set of data. The obtained class probabilities had a range between 0 and 1, where 0 corresponded to a *male-like* brain and 1 to a *female-like* brain. To assess each model's performance, we computed the area under the receiving operating characteristic curves (AUC-ROC curves). To assess the external validation of our models in independent unseen data, we applied the trained models to the QTAB and ABCD data and stored the resulting class probabilities for further analyses. Although the QTAB dataset included genetically related participants (twins and siblings), the independence of the training sample ensured independent sex predictions in the validation samples.

## Statistics and reproducibility

**Biological associations with age and pubertal development**. Since the effects of both age and pubertal development are expected to have opposite direction in males and females (i.e. positive associations in the direction of greater class probabilities in females towards more *female-like brains*, while negative associations in the direction of lower class probabilities in males toward more *male-like brains*), and a significant sex difference is expected in the class probabilities due to the classification model itself, all the analyses were carried out separately for each sex. For each set of analysis, three associations were run for each dataset, one for each machine learning model (limbic, non-limbic and whole brain).

The association with age was investigated in both the training and the two independent test samples. In the training data, a linear model for the class probabilities derived from each of the feature set was implemented, as follows:

$$lm(Class\ Probabilities \sim age_{months} + Euler + site).$$

In QTAB and ABCD, linear mixed effects (LME) models were run, to take into account the longitudinal nature of the data. For this we used the *lme* function of the *nlme* (version 3.1-165) package available in R, including age (expressed in months), session and the interaction between the two as the main independent variables, and to further control our analysis for Euler number. To capture the effects due to repeated measures, we included the subject as random effects in the model. The final model was written as follows:

$$lme\,(Class\ Probabilities \sim age_{months} + session + Euler \\ + age_{months} : session, random =\sim 1|subject)$$

Since the ABCD has multiple study sites, we included the site as further covariate, after excluding sites with less than 20 observations available.

The associations with pubertal development were investigated using the QTAB and ABCD data. To do so, we used the average Youth Self-Report Pubertal Development Scale (PDS) score. This questionnaire consists of 5 questions, two of which are sex-specific, rating different aspects of pubertal development. Three common questions rate the growth in height, the body hair and change in skin. The two sex-specific questions refer in males to facial hair growth and voice break, while in females they refer to breast development and the beginning of menstruation. Each question is rated with a score from 1 to 4 as following: 1 = "Has not yet started", 2 = "Has barely started", 3 = "Is definitely underway" and 4 = "It is completed". The question on menstruation in females constitutes the only exception and it is rated as binary, with "No" = 1 and "Yes" = 4. A caregiver report was also available and was used as further control for supplementary analyses in the ABCD sample.

We first assessed the direct association of class probabilities with the PDS score. To allow the direct comparisons of result between females and males, for each participant the average score of PDS question was calculated at each session and used as variable for the LME models. Each of the LME models for each dataset and set of features included the average PDS score, age (expressed in months) and the interaction between the two, as well as the Euler number as further covariates. Since the PDS average score is expected to increase continuously with age independently from the study session, session was not included in this analysis. The possible correlation between observations due to the longitudinal nature of the data was captured adding a random effect due to subject, as follows:

$$lme(ClassProbabilities \sim PDS_{average} + age_{months} + Euler \\ + PDS_{average} : age_{months}, random =\sim 1|subject).$$

For the ABCD, site was included as a further covariate, after excluding sites with less than 20 observations available.

We applied an ANOVA type I to the output of each LME model, to investigate the main effect of PDS score. As further control and to disentangle the relative contributions of the covariates, we ran an ANOVA type II. The results of these analyses are presented in the Supplementary File.

Next, we assessed the association with menarche onset in both datasets. For this, we divided the females into two groups based on their answers to the PDS question about menarche: 1- females subjects that did not start to menstruate at any of the two sessions of the studies (answered 1 for both session the relative PDS question); 2- females that had menarche (i.e. first menstrual cycle) between the two sessions (answered 1 at baseline and 4 at follow-up). The final sample for QTAB included a total of $N = 110$ participants at baseline (of which $n = 46$ participants had menarche between the two sessions), and $N = 1091$ at baseline for ABCD (of which $n = 414$ participants had menarche between sessions). We then ran an LME model for each machine learning model, similarly to the one described for PDS score, including age (expressed in months), session, the interaction between age and menarche, the interaction between session and menarche and Euler number, capturing the random effect due to repeated measure in the same subject. In the ABCD, we further added as covariate the site, after removing sites with less than 20 observations. The final model was written as follows:

$$lme\,(Class\ Probabilities \sim Menarche\ onset + age_{months} + session + Euler + menarche\ onset\ age_{months} \\ + menarche\ onset : session, random =\sim 1|subject)$$

We applied an ANOVA type I to the output of LME to assess the main effect of menarche onset. We further tested the relative contributions of menarche, age and session by applying an ANOVA type II (Supplementary File).

**Mental health associations**. To investigate the association with mental health in the QTAB sample, we used data from the Spence Children's Anxiety Scale (SCAS), the Short Mood and Feeling Questionnaire (SMFQ) and the anxiety and depression subscale of the Somatic and Psychological Health Report (SPHERE-21) obtained from the self-reported questionnaires. These questionnaires measure symptoms of anxiety and depression and were collected at both sessions, allowing us to assess longitudinal changes in mental health scores. To obtain a unique continuous score, we ran a principal component analysis (PCA) for dimensionality reduction across all questions at baseline and took the first component as a general score for mental health. To assess the mental health score at follow-up, we took the eigenvectors of each question for the first component at baseline and applied them to the data at follow-up, obtaining the PCA scores.

To assess the association of mental health scores with class probabilities, we run an LME model for each machine learning model as following:

$$lme\,(ClassProbabilities \sim PCA\;scores + age_{months} + session + Euler + PCA\;scores :$$
$$age_{months} + PCA\;scores : session, random = \sim 1|subject)$$

We tested for significance of the main effect of mental health by applying an ANOVA type I to the LME output. Further, we tested relative contributions of covariates with an ANOVA type II (see Supplementary File for results).

### Reporting summary

Further information on research design is available in the Nature Portfolio Reporting Summary linked to this article.

### Data and materials availability

All data used to reach the conclusions presented in this paper are presented in the main text or the supplementary material. All raw data is available via dedicated data use agreements with the data providers (see below).We used data from the Philadelphia Neurodevelopmental Cohort (PNC, access permission no 29782), the Human Connectome Project Lifespan—Development (HCP-D), the Adolescent Brain Cognitive Development Study® (ABCD, https://abcdstudy.org), and the Queensland Twin Adolescent Brain Project (QTAB, https://openneuro.org/datasets/ds004146/versions/1.0.4).

Support for the collection of the data for PNC was provided by grant RC2MH089983 awarded to Raquel Gur and RC2MH089924 awarded to Hakon Hakonarson. Subjects were recruited and genotyped through the Center for Applied Genomics (CAG) at The Children's Hospital in Philadelphia (CHOP). Phenotypic data collection occurred at the CAG/CHOP and at the Brain Behavior Laboratory, University of Pennsylvania.

For the HCP-D, data were provided [in part] by the Human Connectome Project, WU-Minn Consortium (Principal Investigators: David Van Essen and Kamil Ugurbil; 1U54MH091657) funded by the 16 NIH Institutes and Centers that support the NIH Blueprint for Neuroscience Research; and by the McDonnell Center for Systems Neuroscience at Washington University.

The ABCD Study® (https://abcdstudy.org), held in the NIMH Data Archive (NDA), is a multisite, longitudinal study designed to recruit more than 10,000 children age 9-10 and follow them over 10 years into early adulthood. The ABCD Study® is supported by the National Institutes of Health and additional federal partners under award numbers U01DA041048, U01DA050989, U01DA051016, U01DA041022, U01DA051018, U01DA051037, U01DA050987, U01DA041174, U01DA041106, U01DA041117, U01DA041028, U01DA041134, U01DA050988, U01DA051039, U01DA041156, U01DA041025, U01DA041120, U01DA051038, U01DA041148, U01DA041093, U01DA041089, U24DA041123, U24DA041147. A full list of supporters is available at https://abcdstudy.org/federal-partners.html. A listing of participating sites and a complete listing of the study investigators can be found at https://abcdstudy.org/consortium_members/. ABCD consortium investigators designed and implemented the study and/or provided data but did not necessarily participate in the analysis or writing of this report. This manuscript reflects the views of the authors and may not reflect the opinions or views of the NIH or ABCD consortium investigators. The ABCD data repository grows and changes over time. The ABCD data used in this report came from https://doi.org/10.15154/zk5y-pc91.

The QTAB project resource was produced as a result of i) the goodwill and contribution of 422 twin/triplet participants and their parents, ii) funding from the National Health and Medical Research Council, Australia (APP1078756) and the Queensland Brain Institute, University of Queensland, iii) access to several key resources, including the Centre for Advanced Imaging, the Human Studies Unit, Institute of Molecular Bioscience, and the Queensland Cyber Infrastructure Foundation, at the University of Queensland, local and national twin registries at the QIMR Berghofer Medical Research Institute and Twin Research Australia, as well as the many assessments made available by researchers worldwide, and iv) was established with the purpose of promoting the conduct of health-related research in adolescence. Access to the data was guaranteed under data user agreement provided at https://doi.org/10.5281/zenodo.7779769.

### Code availability

All code will be made publicly available at https://github.com/gloriamatte upon publication.

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

## Acknowledgements
This work was performed as part of the International Research Training Group: Women's Mental Health Across the Reproductive Years (IRTG 2804). The study was supported by the BMBF-funded de.NBI Cloud within the German Network for Bioinformatics Infrastructure (de.NBI) (031A537B, 031A533A, 031A538A, 031A533B, 031A535A, 031A537C, 031A534A, 031A532B). T.K. received funding from the Interfaculty Graduate Program AI4Med-BW and Fortüne Program (2660-0-0) from the Faculty of Medicine at University of Tübingen, the German Research Foundation (IRTG 2804), the Research Council of Norway (#323961) and the European Research Council (ERC CoG, #101086793, HealthyMom). T.K. is a member of the Machine Learning Cluster of Excellence, EXC number 2064/1 — Project number 39072764. BD received funding from the German Research Foundation (IRTG 2804, DE2319/9-1). E.C. received funding from SciLifeLab. We acknowledge support from the Open Access Publication Fund of the University of Tübingen.

## Author contributions
Conceptualization: G.M.B. and T.K. Methodology: G.M.B. and T.K. Formal analysis: G.M.B. Data curation: G.M.B. and T.K. Data interpretation: G.M.B., J.W., and T.K. Visualization: G.M.B. Writing—Original Draft: G.M.B., and T.K. Writing—Review and Editing: G.M.B., J.W., E.C., B.D., and T.K. Supervision: T.K., E.C., and B.D. Funding Acquisition: T.K., E.C., and B.D.

## Funding

## Competing interests
The authors declare no competing interests.
