## [Transparent Peer Review file · Communications Biology]

Longitudinal development of sex differences in the limbic system is associated with age, puberty and mental health

Corresponding Author: Ms Gloria Matte Bon

Version 0:

Reviewer comments:

Reviewer #1

(Remarks to the Author)

In this manuscript, Matte Bon and colleagues investigated the role of age, puberty, and mental health in shaping the longitudinal development of brain structures. Their analyses revealed several interesting results. First, they showed that sex differences in the brain increase with age. Second, they found that the associations between brain structure and sex (i.e., the sex class probabilities) were linked to age and puberty in a sex-specific manner, with stronger relationships in females. Third, they showed that, in females, associations between limbic structures and sex (i.e., sex class probabilities from the limbic models) were linked to the onset of menarche. Finally, they demonstrated that, in females, associations between brain structures and sex (i.e., sex class probabilities) were linked to mental health. Overall, the analyses are novel, the manuscript is well-written, and the conclusions are justified. Authors also validate their findings using different PDS scores and matching strategies. Therefore, this work is likely to be of interest to Communications Biology readers. However, I do have a few suggestions for how this manuscript could be improved. Please find them below. I am signing my review and am happy to be contacted by the authors if they have any questions.

- Elvisha Dhamala

My main concern is the lack of exploration of which limbic and non-limbic structures are driving the predictions and underlying the relationships being captured here. The authors used a parcellation that includes 160 limbic features and 333 non-limbic features for a total of 493 features, and these features will have differentially contributed to the sex class probabilities that were generated. An examination of the features that were most strongly implicated would be helpful to contextualise the results and gain a deeper understanding of which limbic and non-limbic structures exhibit the strongest relationships. I would also be curious to see whether the limbic/non-limbic and the whole brain models overlap in terms of the feature contributions. In other words, are the feature contributions of non-limbic structures similar when they are used in the non-limbic and the whole brain models? If so, this would demonstrate that the results that are being captured are robust.

Other minor concerns are below:

- A visualisation of the limbic and non-limbic parcellation (projected onto the brain) would be helpful to provide context into the analyses
- Resolution of the figures in the supplementary materials are rather low

Reviewer #2

(Remarks to the Author)

This longitudinal study examines sex differences in brain structure and explores how age, puberty, and mental health associate with these differences. The authors trained machine learning models on cross-sectional neuroimaging data from two cohorts and tested their models on two independent longitudinal developmental neuroimaging cohorts. They found regional sex differences in brain structure (adjusted for total intracranial volume), which became more pronounced with age. Notably, puberty and mental health showed stronger associations with sex-related brain differentiation in females—particularly in limbic regions—than in males, where the effects were absent or less pronounced. Overall, this study addresses a timely and important topic, presenting novel findings that are highly relevant to sex-specific approaches in mental health research. We were impressed by the thoroughness and rigor of the analyses. Below, we offer a few minor suggestions aimed at further enhancing the clarity and impact of this valuable manuscript.

1. INTRODUCTION: The hypotheses that sex differences will increase with age needs more explanation. Sex differences

may increase during puberty, but what happens after puberty? For some of the included individuals puberty ended between baseline and follow-up measurements. Do the authors still expect larger sex differences at follow-up for these individuals and a 'linear' increase of sex differences? We would recommend expanding the introduction with more targeted literature and theoretical grounding to better justify the hypotheses and clarify how they were derived.

2. INTRODUCTION: The concept of a "male-like" to "female-like" brain continuum is interesting and well-motivated, but we recommend clarifying that these estimates reflect biological sex differences and not gender identity or expression. Even though the article uses sex assigned at birth as the reference, terms like "male-like brain" may be misread as implying gender-related traits or identities, especially by broader audiences. A brief clarification would strengthen the interpretation and prevent confusion.

3. METHODS: The QTAB dataset includes adolescent twins, but we could not find any information on how the analyses accounted for the fact that twins are not independent observations. Please clarify whether and how this was handled.

METHODS: The methods should be more specific in the description of the number of participants that were excluded in each step and the justification for these exclusions. Please add a flowchart of the numbers of individuals excluded in each step and the reasons. Also, it would be interesting to address the homogeneity of the final subsamples more in the discussion. The next points are also related to the justification of excluding participants. We understand that it takes some work to add these descriptions with all the different samples, but we feel this will enhance the transparency and reproducibility of the manuscript a lot and will make it easier to interpret the generalizability of the findings.

o We agree that TIV correction is essential and necessary, but it also comes with disadvantages related to representativeness of the samples as the authors describe in the discussion. It would be valuable to be transparent about the participants that were not included in the analyses. Figure S11 shows that large groups of individuals were excluded during the matching procedure (around 50% of the population?). Please add more information (a table might be easiest) on the number of participants, the average age + standard deviation and the average TIV + standard deviation in the excluded and included groups separated by sex.

o The Euler number is used for MRI QC and included as a covariate in the models. Please explain in the method what Euler is and justify why it is needed to exclude individuals based on their Euler number as these exclusions also reduce variation in the datasets.

o The information on the MRI QC is very limited and it is unclear what kind of artefacts were found in the excluded scans.

4. METHODS: Could the improved performance at follow-up be related to less noisy data in older individuals? MRI data in younger individuals could be noisier due to more motion artefacts and/or less optimal fit with the adult template brains used by FreeSurfer. Please explain in the methods if and how these issues were addressed.

5. METHODS: The manuscript has a strong focus on the limbic regions, but it remains unclear which regions were included in this measure. Please list or visualise the regions that are included in the limbic model so future studies can build on the promising results of the current study.

6. METHODS: It is unclear if the structural MRI data were defaced. This is important to know as defacing can impact structural brain measures in an age-dependent manner. Also, if the data was defaced, did the authors visually check the results of the defacing procedures?

7. RESULTS: Line 152: Since the effect of age is expected to go in the opposite direction for each sex (i.e. increase in class probabilities in females in the direction of more female-like brains, while decrease toward more male-like brains in males when increasing age), we carried out the analysis within each sex separately. This sentence is somewhat confusing. Please rephrase it to make clear that the effect is in the opposite direction because of the way the female- and male-like brains were coded rather than this being a 'biological phenomenon'.

8. RESULTS: "The main analysis was conducted using LME in combination with ANOVA type I, assessing the main effect of mental health. In females, we observed significant associations of mental health with the whole brain and limbic class probabilities (although the latter were fairly close to correction threshold at $p = .0167$)." The authors here suggest that p-values are meaningful beyond their use as a cut-off. The sentence between brackets appears to confuse p-value and effect size. We recommend removing the sentence between brackets to avoid confusion and just stick to the a priori decided threshold.

Reviewer #3

(Remarks to the Author)

I co-reviewed this manuscript with one of the reviewers who provided the listed reports. This is part of the Communications Biology initiative to facilitate training in peer review and to provide appropriate recognition for Early Career Researchers who co-review manuscripts.

Version 1:

Reviewer comments:

Reviewer #1

(Remarks to the Author)

Authors have addressed all of my concerns and I am excited to see this work published.

Reviewer #2

(Remarks to the Author)

Some of our comments were not completely addressed or sufficiently substantiated in the rebuttal letter. Nevertheless, the revisions to the manuscript itself have resolved our main concerns, and we therefore believe it is now suitable for

publication.

Reviewer #3

(Remarks to the Author)

I co-reviewed this manuscript with one of the reviewers who provided the listed reports. This is part of the Communications Biology initiative to facilitate training in peer review and to provide appropriate recognition for Early Career Researchers who co-review manuscripts.

Longitudinal development of sex differences in the limbic system is associated with age, puberty and mental health

COMMSBIO-25-4013-T

Response to all reviewers

We would like to thank the reviewers for their valuable feedback which contributed to improving the manuscript from the initial submission. In brief, we included the following revisions:

- We implemented a new analysis to investigate which features contributed to each model (whole brain, limbic, non-limbic) and subsequently compared these contributions between models.
- We added the list of limbic features to clarify which brain regions were regarded as limbic. Furthermore, we added respective cortical brain maps that visualize the cortical distribution of limbic and non-limbic features.
- Given the various different data sets and analyses used, we added more details on sample composition of each analysis. Specifically, we added a table with the summary statistics for the included and excluded participants in our training sample, and a second table with the number of subjects included for each association analysis in the two longitudinal samples.
- We improved the description of the methods and clarify the terminology to avoid confusion between sex and gender identity.
- We elaborated more on the limitations of our work

Individual, specific responses to reviewers' comments can be found below. We have addressed all of them and have incorporated corresponding changes in the manuscript also directly in the response letter below in quotation marks and italics font. Changes in the revised manuscript are highlighted with **green color**.

Reviewer 1: page 2

Reviewer 2: page 5

Reviewer 3: page 15

Reviewer #1

In this manuscript, Matte Bon and colleagues investigated the role of age, puberty, and mental health in shaping the longitudinal development of brain structures. Their analyses revealed several interesting results. First, they showed that sex differences in the brain increase with age. Second, they found that the associations between brain structure and sex (i.e., the sex class probabilities) were linked to age and puberty in a sex-specific manner, with stronger relationships in females. Third, they showed that, in females, associations between limbic structures and sex (i.e., sex class probabilities from the limbic models) were linked to the onset of menarche. Finally, they demonstrated that, in females, associations between brain structures and sex (i.e., sex class probabilities) were linked to mental health. Overall, the analyses are novel, the manuscript is well-written, and the conclusions are justified. Authors also validate their findings using different PDS scores and matching strategies. Therefore, this work is likely to be of interest to Communications Biology readers. However, I do have a few suggestions for how this manuscript could be improved. Please find them below. I am signing my review and am happy to be contacted by the authors if they have any questions.

- Elvisha Dhamala

We would like to thank the reviewer for the positive feedback and the constructive comments that have helped us improve our work.

1. My main concern is the lack of exploration of which limbic and non-limbic structures are driving the predictions and underlying the relationships being captured here. The authors used a parcellation that includes 160 limbic features and 333 non-limbic features for a total of 493 features, and these features will have differentially contributed to the sex class probabilities that were generated. An examination of the features that were most strongly implicated would be helpful to contextualise the results and gain a deeper understanding of which limbic and non-limbic structures exhibit the strongest relationships. I would also be curious to see whether the limbic/non-limbic and the whole brain models overlap in terms of the feature contributions. In other words, are the feature contributions of non-limbic structures similar when they are used in the non-limbic and the whole brain models? If so, this would demonstrate that the results that are being captured are robust.

We thank the reviewer for the suggestion and fully agree that a closer look to feature contribution would help to contextualize the findings and strengthen our results. Accordingly, we performed a supplementary analysis to investigate feature contributions across models. Interestingly, we found high correspondence between the whole brain model and each of the two regional models (limbic and non-limbic) in terms of feature contribution. Regions contributing the most to the limbic model were the septal nuclei, the piriform cortex, subregions of the anterior and posterior cingulate cortex and the mammillary body. Regions contributing the most to the non-limbic model were the peri-Sylvian language area and the left medial pulvinar.

Moreover, out of the first ten features contributing to the whole brain model, eight were limbic features. We incorporated these new findings alongside a new supplementary figure into the result section:

“Interestingly, when comparing the feature contributions in the two regionally constrained models to the feature contribution in the whole brain model, we found a considerable overlap between the whole brain and respective regional models. Similar features contributed to the regional and whole brain models, indicating that the model predictions are robust (Supplementary Figure S1). Of note, eight out of the first ten features contributing to the whole brain model were limbic features, providing further evidence of the relevance of limbic features to uncover sex-specific factors.”

(p. 6, l. 137-143)

Figure S1. Comparisons of feature contributions between regionally constrained and whole brain models. Features contributing the most to the limbic and non-limbic models also contributed the most to the whole brain model. Regions contributing the most to the limbic model were the septal nuclei, the piriform cortex, subregions of the anterior and posterior cingulate cortex and the mammillary body. Regions contributing the most to the non-limbic model were the peri-Sylvian language area and the left medial pulvinar.

Other minor concerns are below:

- A visualisation of the limbic and non-limbic parcellation (projected onto the brain) would be helpful to provide context into the analyses

We agree with the reviewer, and we added the visualization of the cortical limbic and non-limbic structure in Supplementary Figure S13, pasted below. Moreover, we provided the full list of limbic structures and correspondent parcellation used for each cortical and subcortical region in Supplementary Table T3. We added the following text to the methods:

“A complete list of limbic features and respective parcellation is listed in the Supplementary table T3, while Supplementary figure S13 illustrates the cortical distribution of limbic and non-limbic features.”

(p. 24, l. 498-500)

Figure S13. Cortical distribution of limbic and non-limbic features.

- Resolution of the figures in the supplementary materials are rather low

We apologize for the inconvenience. We increased the resolution of the figures.

Reviewer #2

This longitudinal study examines sex differences in brain structure and explores how age, puberty, and mental health associate with these differences. The authors trained machine learning models on cross-sectional neuroimaging data from two cohorts and tested their models on two independent longitudinal developmental neuroimaging cohorts. They found regional sex differences in brain structure (adjusted for total intracranial volume), which became more pronounced with age. Notably, puberty and mental health showed stronger associations with sex-related brain differentiation in females—particularly in limbic regions—than in males, where the effects were absent or less pronounced. Overall, this study addresses a timely and important topic, presenting novel findings that are highly relevant to sex-specific approaches in mental health research. We were impressed by the thoroughness and rigor of the analyses. Below, we offer a few minor suggestions aimed at further enhancing the clarity and impact of this valuable manuscript.

We thank the reviewer for the careful evaluation of our work and for the constructive comments that helped us to improve the paper.

1. INTRODUCTION: The hypotheses that sex differences will increase with age needs more explanation. Sex differences may increase during puberty, but what happens after puberty? For some of the included individuals puberty ended between baseline and follow-up measurements. Do the authors still expect larger sex differences at follow-up for these individuals and a ‘linear’ increase of sex differences? We would recommend expanding the introduction with more targeted literature and theoretical grounding to better justify the hypotheses and clarify how they were derived.

We agree that the theoretical background for this hypothesis was not well addressed. Our reasoning was that brain development is a long-lasting process that extends from early life until adulthood. While we expect pubertal development to be the main driver of sex differences and thus expect strong neural alterations around puberty, the development of sex differences is likely to be characterized by other factors as well, such as for example social factors that span beyond puberty. Therefore, age and puberty are highly intertwined but one might expect to see also age-dependent manifestation of sex differences that is not directly linked to the biological underpinnings of puberty. Unfortunately, the specific effects of age and puberty on the development of sex differences are still largely understudied, with most studies focusing either on age or pubertal effects, but not in their conjunction. To further clarify this point we have modified the introduction as follow:

“Brain development is a long-lasting process that starts in early life and continues until adulthood, characterized by the interplay of genetic, hormonal and social

factors. In this framework, age and puberty, highly interconnected, might interact in establishing sex differences in the brain, by differentially affecting the temporal dynamics of brain development across sexes (20, 21). While certain manifestations of sex differences in the brain may be directly attributable to the biological processes underlying pubertal development, others may be attributable to age dependent social or genetic factors that go beyond puberty. As such, disentangling the effects of age and puberty is crucial to fully understand brain dynamics across adolescence.”

(p. 3, l. 66-74)

[20] M. Curtis, J. C. Flournoy, S. Kandala, A. F. P. Sanders, M. P. Harms, A. Omary, L. H. Somerville, D. M. Barch, Disentangling the unique contributions of age, pubertal stage, and pubertal hormones to brain structure in childhood and adolescence. *Dev. Cogn. Neurosci.* 70, 101473 (2024).

[21] M. C. Holm, E. H. Leonardsen, D. Beck, A. Dahl, R. Kjelkenes, A.-M. G. De Lange, L. T. Westlye, Linking brain maturation and puberty during early adolescence using longitudinal brain age prediction in the ABCD cohort. *Dev. Cogn. Neurosci.* 60, 101220 (2023).

We further clarify the expected effect of age based on the literature on brain sex during development in the following paragraph:

“Kurth and colleagues (5) used structural imaging phenotypes to investigate changes in brain sex classification accuracy in association with age across childhood and adolescence, showing better performance for adolescents compared to children and increased magnitude of sex differences with increasing age.”

(p. 4, l. 90-91)

[5] F. Kurth, C. Gaser, E. Luders, Development of sex differences in the human brain. *Cogn. Neurosci.* 12, 155–162 (2021).

2. INTRODUCTION: The concept of a "male-like" to "female-like" brain continuum is interesting and well-motivated, but we recommend clarifying that these estimates reflect biological sex differences and not gender identity or expression. Even though the article uses sex assigned at birth as the reference, terms like “male-like brain” may be misread as implying gender-related traits or identities, especially by broader audiences. A brief clarification would strengthen the interpretation and prevent confusion.

We thank the reviewers for pointing out the risk of misinterpretation of such phrasing. We fully agree that the current phrasing might be misleading. We therefore added clarification in the introduction:

“Importantly, such position is not an expression of gender identity, but rather a manifestation of the overall pattern of sex differences across the brain regions under consideration.”

(p. 4, l. 83-85)

3. METHODS: The QTAB dataset includes adolescent twins, but we could not find any information on how the analyses accounted for the fact that twins are not independent observations. Please clarify whether and how this was handled.

We apologize for the lack of information concerning the handling of twin subjects. We used fully independent samples for model training and external validation. QTAB was solely used in validation, and the respective predictions work at the single subject level. Therefore, the trained models predict sex independently for each observation in the validation sample, without assuming any relation between data. Therefore, at the prediction stage there is no impact of twinship confounds. However, we fully agree that at the level of statistical associations these relationships can play a role and that we cannot rule out the influence of shared genetic factors on our results. In fact, in our previous work we have established that class probabilities from similar models are heritable. Unfortunately, such analyses would go beyond the scope of the current work and we therefore decided to follow this up in a dedicated study but here discuss these as a limitation instead:

“Finally, the QTAB samples include twin and sibling participants. For model validation this is not an issue per se as we ensured independence of sex predictions between subjects by predicting at the individual subject level from a model trained in other datasets. However, we did not account for family factors in our LME models when testing for associations of the class probabilities. In light of our previous findings in adults, showing high degree of heritability for class probabilities (27), further research on the impact of kinship in the observed associations might be warranted as we cannot currently quantify the impact of shared genetic factors between siblings.”

(p. 21, l. 420-427)

[27] G. Matte Bon, D. Kraft, E. Comasco, B. Derntl, T. Kaufmann, Modeling brain sex in the limbic system as phenotype for female-prevalent mental disorders. *Biol. Sex Differ.* 15, 42 (2024).

Moreover, we further specified in the methods section the independence of prediction in the validation samples:

“Although the QTAB dataset included genetically related participants (twins and siblings), the independence of the training sample ensured independent sex predictions in the validation samples.”

(p. 25, l. 516-518)

4. METHODS: The methods should be more specific in the description of the number of participants that were excluded in each step and the justification for these

exclusions. Please add a flowchart of the numbers of individuals excluded in each step and the reasons. Also, it would be interesting to address the homogeneity of the final subsamples more in the discussion. The next points are also related to the justification of excluding participants. We understand that it takes some work to add these descriptions with all the different samples, but we feel this will enhance the transparency and reproducibility of the manuscript a lot and will make it easier to interpret the generalizability of the findings.

We thank the reviewer for pointing out the lack of clarity concerning the sample description. While for the training sample the exclusion of subjects depended on quality control and matching procedure, for the two validation samples after exclusion for quality control as reported in the text, the sample size for each analysis was determined by data availability at each time point. To clarify this point we added two supplementary tables, one for the training sample (see next point), and one for the two validation samples, pasted here for convenience together with the corresponding text in the main manuscript, reporting the mean age and standard deviation at each timepoint for each session and the number of subjects included in each analysis:

“Supplementary Table T2 reports the included participants for each analysis.”
(p. 23, l. 469)

			Baseline		Follow-up	
			Females	Males	Females	Males
QTAB						
	Age		137 ± 16.87	136 ± 15.79	158 ± 19.14	155 ± 17.54
	N					
		Tot	192	200	152	138
		PDS	185	179	145	119
		Menarche	110	-	110	-
		No menarche	64	-	64	-
		Between sessions	46	-	46	-
		Mental Health	192	200	152	138
ABCD						
	Age		119 ± 7.41	119 ± 7.49	143 ± 7.78	144 ± 7.75
	N					
		Tot	3629	4121	3478	4070
		PDS	1173	1846	1159	1841
		Menarche	1091	-	1091	-
		No menarche	677	-	677	-
		Between sessions	414	-	414	-

Table T2. Included participants for each association analysis in the two longitudinal samples. Age is expressed in months and reported as mean ± standard deviation (SD)

We decided, however, against a flowchart for the matching procedure, as this would be a repetition of the steps reported in the text and metrics reported in the supplementary table and Figure S12. We believe that the combination of our detailed text with the newly added table provides full details on all aspects of sample composition.

- a) We agree that TIV correction is essential and necessary, but it also comes with disadvantages related to representativeness of the samples as the authors describe in the discussion. It would be valuable to be transparent about the participants that were not included in the analyses. Figure S11 shows that large groups of individuals were excluded during the matching procedure (around 50% of the population?). Please add more information (a table might be easiest) on the number of participants, the average age + standard deviation and the average TIV + standard deviation in the excluded and included groups separated by sex.

We thank the reviewer for the thoughtful comment and agree that there is a trade-off between sample bias and representativeness. We added a supplementary table for the training sample, pasted for convenience below, and added the reference text in the methods. The comparisons between total sample, selected sample and excluded sample regarding their mean \pm sd eTIV shows (1) that the matching worked well in terms of removing bias (eTIV of males and females are well aligned in the selected sample) but also (2) that the resulting values are not completely unrepresentative, as they align well with the values observed in the total distribution (e.g. mean of the selected male sample well within the bounds of 1 SD from the mean of the total sample).

“Supplementary Table T1 shows the sample characteristic before and after the matching procedure for the selected and excluded participants.”

(p.23, l. 457-458)

	Total Sample		
	Tot	Females	Males
N	2054	1086	968
age	179 ± 44.5	181 ± 45.0	178 ± 43.9
eTIV	1436981 ± 154645.9	1361084 ± 136872.0	1522131 ± 126595.0
	Selected		
	Tot	Females	Males
N	1132	566	566
age	174 ± 44.6	174 ± 44.5	174 ± 44.7
eTIV	1449042 ± 98151.0	1447216 ± 99663.0	1450869 ± 96670.0
	Excluded		
	Tot	Females	Males
N	922	520	402
age	187 ± 43.4	188 ± 44.3	184 ± 42.0
eTIV	1422173 ± 202685.0	1267332 ± 107026.0	1622465 ± 90666.0

Table T1. Training sample characteristics before and after the matching procedure for the selected and excluded participants. The eTIV distributions support that the matching procedure successfully removed the eTIV difference between males and females, while still staying within the bounds of a reasonably representative sample. Mean ± standard deviation (SD) of age (expressed in months) and eTIV.

b) The Euler number is used for MRI QC and included as a covariate in the models. Please explain in the method what Euler is and justify why it is needed to exclude individuals based on their Euler number as these exclusions also reduce variation in the datasets.

We apologize for the lack of details on MRI quality control and the rationale behind excluding subjects based on the Euler number. The Euler number is a FreeSurfer derived measure of the topological complexity of the cortical reconstruction, and it has been shown to correlate with manual rating procedure of image quality evaluation and parameters such as motion artifacts. We extended the Euler number description by providing further details on the Euler number and its relation to image quality, as follows:

“As quality control of the imaging data, we used the Euler number, a proxy of image data quality (35), averaged across the two hemispheres and excluded subjects with values lower than three standard deviations from the mean. The Euler number represents a measure of the topological complexity of the reconstructed cortex and has been shown to provide a useful proxy of structural image quality, given high association with manual rating procedures and outperforming motion indicators in identifying unusable images (35, 62). By excluding outliers for Euler number in each

sample we aim to control for MRI quality reconstruction and potential motion artefacts.”

(p. 24, l. 481-488)

[35] A. F. G. Rosen, D. R. Roalf, K. Ruparel, J. Blake, K. Seelaus, L. P. Villa, R. Ciric, P. A. Cook, C. Davatzikos, M. A. Elliott, A. Garcia de La Garza, E. D. Gennatas, M. Quarmley, J. E. Schmitt, R. T. Shinohara, M. D. Tisdall, R. C. Craddock, R. E. Gur, R. C. Gur, T. D. Satterthwaite, Quantitative assessment of structural image quality. *NeuroImage* 169, 407–418 (2018).

[62] T. Kaufmann, D. van der Meer, N. T. Doan, E. Schwarz, M. J. Lund, I. Agartz, D. Alnæs, D. M. Barch, R. Baur-Streubel, A. Bertolino, F. Bettella, M. K. Beyer, E. Bøen, S. Borgwardt, C. L. Brandt, J. Buitelaar, E. G. Celius, S. Cervenka, A. Conzelmann, A. Córdova-Palomera, A. M. Dale, D. J. F. de Quervain, P. Di Carlo, S. Djurovic, E. S. Dørum, S. Eisenacher, T. Elvsåshagen, T. Espeseth, H. Fatouros-Bergman, L. Flyckt, B. Franke, O. Frei, B. Haatveit, A. K. Håberg, H. F. Harbo, C. A. Hartman, D. Heslenfeld, P. J. Hoekstra, E. A. Høgestøl, T. L. Jernigan, R. Jonassen, E. G. Jönsson, P. Kirsch, I. Kłoszewska, K. K. Kolskår, N. I. Landrø, S. Le Hellard, K.-P. Lesch, S. Lovestone, A. Lundervold, A. J. Lundervold, L. A. Maglanoc, U. F. Malt, P. Mecocci, I. Melle, A. Meyer-Lindenberg, T. Moberget, L. B. Norbom, J. E. Nordvik, L. Nyberg, J. Oosterlaan, M. Papalino, A. Papassotiropoulos, P. Pauli, G. Pergola, K. Persson, G. Richard, J. Rokicki, A.-M. Sanders, G. Selbæk, A. A. Shadrin, O. B. Smeland, H. Soininen, P. Sowa, V. M. Steen, M. Tsolaki, K. M. Ulrichsen, B. Vellas, L. Wang, E. Westman, G. C. Ziegler, M. Zink, O. A. Andreassen, L. T. Westlye, Common brain disorders are associated with heritable patterns of apparent aging of the brain. *Nat. Neurosci.* 22, 1617–1623 (2019).

c) The information on the MRI QC is very limited and it is unclear what kind of artefacts were found in the excluded scans.

We thank the reviewer for pointing out the lack of information on MRI quality control. This has been addressed through improved descriptions as described in the point above.

5. METHODS: Could the improved performance at follow-up be related to less noisy data in older individuals? MRI data in younger individuals could be noisier due to more motion artefacts and/or less optimal fit with the adult template brains used by FreeSurfer. Please explain in the methods if and how these issues were addressed.

We agree with the reviewers that motion artefacts could affect our results. However, by excluding subjects based on the Euler number, we aim to control for such artifacts. The Euler number has been shown to outperform indicators of motion in identifying unusable images. Moreover, while motion artifacts can cause tremendous issues for timeseries data obtained with functional MRI, their effect on structural MRI data is expected to be much lower. Therefore, we considered it appropriate to base our MRI QC exclusion criteria on the Euler number, in line with established procedures in our earlier work. To clarify this point, we explained the

relation between motion indicators and Euler number in the description of quality control, as already reported in point 4b:

“The Euler number represents a measure of the topological complexity of the reconstructed cortex and has been shown to provide a useful proxy of structural image quality, given high association with manual rating procedures and outperforming motion indicators in identifying unusable images (35, 62).”
(p. 24, l. 483-486)

6. METHODS: The manuscript has a strong focus on the limbic regions, but it remains unclear which regions were included in this measure. Please list or visualise the regions that are included in the limbic model so future studies can build on the promising results of the current study.

We fully agree with the reviewer that a visualization and list of limbic features will help further replication of the study. This point was also brought up by reviewer #1. Accordingly, we added a supplementary figure to show the cortical distribution of limbic and non-limbic features and further added a supplementary table (Supplementary Table T3) containing the full list of limbic features, the segmentation approach used and the corresponding name in FreeSurfer. We added the corresponding text reference for the supplementary figure and table in the method section:

“A complete list of limbic features and respective parcellation is listed in the Supplementary table T3, while Supplementary figure S13 illustrates the cortical distribution of limbic and non-limbic features.”
(p. 24, l. 498-500)

Figure S13. Cortical distribution of limbic and non-limbic features.

7. METHODS: It is unclear if the structural MRI data were defaced. This is important to know as defacing can impact structural brain measures in an age-dependent manner. Also, if the data was defaced, did the authors visually check the results of the defacing procedures?

We thank the reviewer for pointing out this issue. We retrieved the data from external sources and some of these have been defaced while others have not. Specifically, the data used in training samples were a combination of non-defaced (PNC) and defaced (HCP-D) data. Likewise in the validation samples, QTAB data were defaced while ABCD data were not. We agree with the reviewers that defacing can impact the quality of structural brain measurement derivation. However, because we did not have access to the un-defaced data for some samples, we could not implement a procedure to avoid potential de-facing related issues. All that said, we would like to argue that the impact of defacing on our current findings is likely minor. By combining defaced and non-defaced data in the training sample and by independently testing the models on defaced and non-defaced data, it is unlikely that defacing issues could explain the results, if the results are consistent across samples, as reported in this study. However, we acknowledge the limitation, and have therefore modified the methods section to include the description of defaced and non-defaced samples, as follows:

“A mix of defaced and non-defaced data was used for both the training (PNC not defaced, HCP-D defaced) and the two independent test samples (QTAB defaced, ABCD not defaced), as provided by the study sources. While defacing may impact brain measures, alignment of results across defaced and not defaced samples supports that findings are robust to potential defacing related issues.”
(p.23, l. 476-480)

8. RESULTS: Line 152: Since the effect of age is expected to go in the opposite direction for each sex (i.e. increase in class probabilities in females in the direction of more female-like brains, while decrease toward more male-like brains in males when increasing age), we carried out the analysis within each sex separately. This sentence is somewhat confusing. Please rephrase it to make clear that the effect is in the opposite direction because of the way the female- and male-like brains were coded rather than this being a ‘biological phenomenon’.

We apologize for the lack of clarity on the direction of the age effects, and we thank the reviewer for pointing it out. We have now rephrased the sentence as follows:

“Since the better performance with increasing age is expected to manifest as a shift of the class probabilities toward the opposite extreme of the distribution for each sex, resulting in an increase in class probabilities toward more female-like brain for females (binary coded in the model as 1), and a decrease toward more male-like brain for males (coded in the model as 0), we carried out the analysis within each sex separately.”
(p.8, l. 170-174)

9. RESULTS: “The main analysis was conducted using LME in combination with ANOVA type I, assessing the main effect of mental health. In females, we observed significant associations of mental health with the whole brain and limbic class probabilities (although the latter were fairly close to correction threshold at $p = .0167$).” The authors here suggest that p-values are meaningful beyond their use as a cut-off. The sentence between brackets appears to confuse p-value and effect size. We recommend removing the sentence between brackets to avoid confusion and just stick to the a priori decided threshold.

We thank the reviewer for their suggestion, and we have now removed the sentence between brackets from the manuscript to avoid confusion.

Reviewer #3

I co-reviewed this manuscript with one of the reviewers who provided the listed reports. This is part of the Communications Biology initiative to facilitate training in peer review and to provide appropriate recognition for Early Career Researchers who co-review manuscripts.

We thank the reviewer for their role in reviewing the manuscript and for participating in such an initiative. The constructive comments provided by the reviewers helped us to improve the paper.